# Full factorial construction of synthetic microbial communities

**Juan Diaz-Colunga**[1,2†‡], **Pablo Catalan**[3†], **Magdalena San Roman**[1,2†], **Andrea Arrabal**[1,2†], **Alvaro Sanchez**[1,2*†]

[1]Instituto de Biología Funcional y Genómica (IBFG), CSIC and Universidad de Salamanca, Salamanca, Spain; [2]Centro Nacional de Biotecnologia (CNB), CSIC, Madrid, Spain; [3]Grupo Interdisciplinar de Sistemas Complejos (GISC), Departamento de Matemáticas, Universidad Carlos III de Madrid, Leganés, Spain

**\*For correspondence:**
alvaro.sanchez@usal.es

[†]These authors contributed equally to this work

**Present address:** [‡]Instituto de Productos Lácteos de Asturias (IPLA), CSIC, Oviedo, Spain

**Competing interest:** The authors declare that no competing interests exist.

## eLife Assessment

This manuscript introduces a new low-cost and accessible method for assembling combinatorially complete microbial consortia using basic laboratory equipment, which is a **valuable** contribution to the field of microbial ecology and biotechnology. The evidence presented is **compelling**, demonstrating the method's effectiveness through empirical testing on both synthetic colorants and *Pseudomonas aeruginosa* strains.

**Abstract** Constructing combinatorially complete species assemblages is often necessary to dissect the complexity of microbial interactions and to find optimal microbial consortia. At the moment, this is accomplished through either painstaking, labor-intensive liquid handling procedures, or through the use of state-of-the-art microfluidic devices. Here, we present a simple, rapid, low-cost, and highly accessible liquid handling methodology for assembling all possible combinations of a library of microbial strains, which can be implemented with basic laboratory equipment. To demonstrate the usefulness of this methodology, we construct a combinatorially complete set of consortia from a library of eight *Pseudomonas aeruginosa* strains, and empirically measure the community-function landscape of biomass productivity, identify the highest-yield community, and dissect the interactions that lead to its optimal function. This easy-to-implement, inexpensive methodology will make the assembly of combinatorially complete microbial consortia easily accessible for all laboratories.

## Introduction

Microbial consortia are playing an increasingly important role in biotechnology. They possess substantial potential advantages over monocultures, including larger metabolic capabilities, division of labor, and a potentially higher ecological and evolutionary stability (*Brenner et al., 2008*). Synthetic microbial communities are being engineered for goals as diverse as degrading pollutants (*Arias-Sánchez et al., 2024*), producing high-value molecules such as biofuels (*Minty et al., 2013*; *Senne de Oliveira Lino et al., 2021*), vitamins (*Wang et al., 2016*; *Wang et al., 2021*), and flavonoids (*Du et al., 2020*; *Xu et al., 2020*), and preventing the invasion of pathogens (*Oliveira et al., 2024*).

To fully realize the promise of microbial consortia in biotechnology, we must develop tools for their optimization. Given a library of candidate strains, which consortia should we form if we wish to maximize a function of interest? (*Diaz-Colunga et al., 2024*) One way to address this question is purely empirical, following a full factorial design in which we form every possible assemblage and simply select whichever one is best (*Angeles-de Paz et al., 2023*; *Kuebbing et al., 2015*; *Langenheder*

*et al., 2010*; *Purswani et al., 2017*; *Sanchez et al., 2023*). Even for candidate libraries of small size (less than 10 species), the combinatorial nature of this strategy makes the assembly of all possible communities painstaking and tedious, limiting the scale and replicability of full factorial design. The number of unique liquid handling events required to form all possible combinations of $m$ species scales as $m\,2^{m-1}$, as each of the $m$ species needs to be added to each of the $2^{m-1}$ consortia where it is present. Manually making all combinations is therefore draining, slow, and prone to human error. Because of the long time required to complete such a task, the risk of contamination is high, and replication is constrained. Perhaps for this reason, studies reporting full factorial construction of microbial consortia are very rare (*Angeles-de Paz et al., 2023*; *Langenheder et al., 2010*; *Purswani et al., 2017*; *Gould et al., 2018*), and the field has largely relied on fractional factorial design, where only a subset of representative species combinations are constructed (*Diaz-Colunga et al., 2024*; *Clark et al., 2021*; *Eng and Borenstein, 2019*; *Sanchez-Gorostiaga et al., 2019*).

The ability to form full factorial combinations of a library of species would have benefits beyond biotechnology. In ecology, forming every possible species assemblage is required if we wish to investigate the complex, combinatorial nature of microbial interactions (*Gould et al., 2018*; *Baichman-Kass et al., 2023*; *Bergelson et al., 2021*; *Eble et al., 2023*; *Friedman et al., 2017*; *Ludington, 2022*; *Picot et al., 2023*) and the role that high-order interactions play in the function and dynamics of microbial communities (*Sanchez-Gorostiaga et al., 2019*; *Ludington, 2022*; *Ansari et al., 2019*; *Kehe et al., 2019*; *Morin et al., 2022*; *Sanchez, 2019*; *Mickalide and Kuehn, 2019*; *Skwara et al., 2023*). It would also allow for the testing and validation of theoretical and computational models of community function (*Diaz-Colunga et al., 2024*; *Ludington, 2022*; *Picot et al., 2023*; *Skwara et al., 2023*). In sum, the quantitative understanding of high-order microbial interactions, the empirical mapping of community assembly networks, the quantitative exploration of community-function landscapes, and the development of predictive models of microbial community assembly and function all require us to be able to reliably assemble communities of growing complexity (*Sanchez et al., 2023*; *Eng and Borenstein, 2019*; *Skwara et al., 2023*).

Robotic liquid handlers can facilitate the task of assembling a full combinatorial set of microbial communities (*Angeles-de Paz et al., 2023*). While this alleviates some of the challenges associated with combinatorial liquid handling, particularly those associated with human error, others remain: the execution of hundreds or thousands of pipetting events is still inevitably slow, and as a result, additional equipment (i.e. a HEPA filter) is required to avoid contamination. Robotic liquid handlers are also expensive, technically sophisticated equipment, and they are not routinely available. As an alternative to deal with the combinatorial complexity of community assembly, Blainey et al. have introduced a droplet-based microfluidic system (kChip) that is capable of forming hundreds of thousands of species assemblages, (*Kehe et al., 2019*; *Kehe et al., 2021*) facilitating their full factorial construction. Notwithstanding their power and unparalleled high-throughput, microdroplet-based approaches such as kChip and others (*Hsu et al., 2019*) are still state-of-the-art experimental methods, requiring specialized equipment and training that are not yet available to the vast majority of research groups worldwide.

In this paper, we describe a simple, rapid, inexpensive, and highly accessible liquid handling methodology for the full factorial design of microbial consortia and inocula, which requires only basic laboratory equipment and which can be easily implemented by the vast majority of research laboratories. Using a standard multichannel pipette, a single user can manually assemble all possible combinations of up to 10 species in less than one hour, a timescale that is shorter than the replication time of most bacteria in minimal media. Higher-dimensional designs are possible too, with the only limitation being the plasticware that is required. In addition to providing detailed protocols for a specific set of conditions (e.g. volume of the culture, density of the bacteria, etc.), we provide an R script to help the user tailor the protocol to their needs (*Source code 1*). To illustrate the usefulness of this approach, we apply it to empirically characterize the full, combinatorially complete community-function landscape of a model microbial consortium consisting of eight *Pseudomonas aeruginosa* strains. We characterize the full absorption spectrum of all consortia and, as a model function, we determine their total biomass. We show that the implementation of our protocol, enabling the full factorial assembly of all strain combinations, allows us to quantitatively determine the relationship between community diversity and function, identify optimal strain combinations, and characterize all pairwise and higher-order interactions among all members of the consortia.

We expect this methodology will rapidly expand the number of factorially constructed microbial consortia in the literature.

## Materials and methods

In what follows, we describe the reasoning behind our protocol in full detail. Readers interested in the practical implementation of this logic are referred to Appendix 1, where we provide a step-by-step guide for the assembly of all consortia from a particular example, consisting of a library of eight microbial isolates. In addition, we provide an R script (**Source code 1**) which can be used to generate similar step-by-step guides for different numbers of species and pipetting volumes.

### Logic of the assembly protocol

The mathematical basis of our method lies in identifying each microbial consortia by a unique binary number. For a set of $m$ species, we can represent any consortium (which we generically call $c$) as

$$c = x_m x_{m-1} x_{m-2} \cdots x_2 x_1 \tag{1}$$

where $x_k = 0, 1$ represents the absence (0) or presence (1) of species $k$ in the consortium (**Figure 1A**). For example, for $m=6$ species, the consortium where only species 1, 2, and 3 are present would be represented as 000111.

Binary numbers are the natural way to denote combinations in mathematics, and indeed the correspondence between the two sets is often used in combinatorics. For instance, there are $\binom{m}{n}$ combinations of $m$ species. The same number counts how many binary strings of length $m$ contain $n$ ones (equivalently, $n$ zeros).

Now, the most important aspect of this notation is that merging two disjoint consortia becomes a simple addition: if we combine consortium 110000 with consortium 000011, the resulting consortium is represented by the binary number 110011, which is the sum of the binary representations of the initial consortia: 110000+000011 = 110,011. Note, however, that this operation only works for disjoint consortia. If we add two consortia sharing one or more species, binary addition is no longer a good representation: combining consortium 100001 with consortium 000001 should result in the first consortium again, 100001 (we are for now ignoring potential differences in species abundances/densities, but will address this issue later in the text). Yet, the addition of the two binary representations leads to 100010, which does not represent the appropriate consortium (remember that in binary addition 01+01 = 10). As we will see in what follows, our protocol minimizes the number of liquid handling events by making extensive use of this addition property, but only for disjoint consortia.

The second property that we will use to our advantage is that 96-well plates have 8 rows, which is a power of 2 ($2^3=8$). This allows us to form all combinations from a three-species set (say, species 1, 2, and 3) in one column of the plate. It is convenient to arrange these 8 consortia following the order of their binary representation: the empty consortium (000) in the first well, followed by 001, 010, 011, 100, 101, 110, and 111; corresponding to the decimal numbers from 0 to 7 in increasing order.

We can now duplicate these eight consortia, pipetting them into the second column of the 96-well plate, and then pipette species 4 (1000) into all wells of this second column using a multichannel pipette (**Figure 1B**). This operation is equivalent to the binary addition of consortium 1000 (species 4 alone) with each of the starting eight consortia, and results in all $2^4=16$ consortia which can be assembled with species 1–4: in the first column, as explained before, we have 0000, 0001, 0010, 0011, 0100, 0101, 0110, and 0111 (decimal numbers 0–7); while in the second column we now have 1000, 1001, 1010, 1011, 1100, 1101, 1110, and 1111 (decimal numbers 8–15, **Figure 1B**). We can next duplicate each of these 16 consortia, pipetting the first and second columns of the plate into the third and fourth columns, respectively. If we then add species 5 (10000) to each of the duplicated consortia, we generate all $2^5=32$ combinations of species 1–5 (**Figure 1B**).

In what follows, we will describe how this logic can be implemented most efficiently in practice, but hopefully the reader can already intuit how to generalize this algorithm for consortia with more species. Note that for more than 6 species, multiple 96-well plates will be required: for instance, with $m=7$ one could form $2^7=128$ potential consortia, more than what a single 96-well plate can fit. Although we will not make use of them here, the algorithm is also generalizable for 384-well plates, as

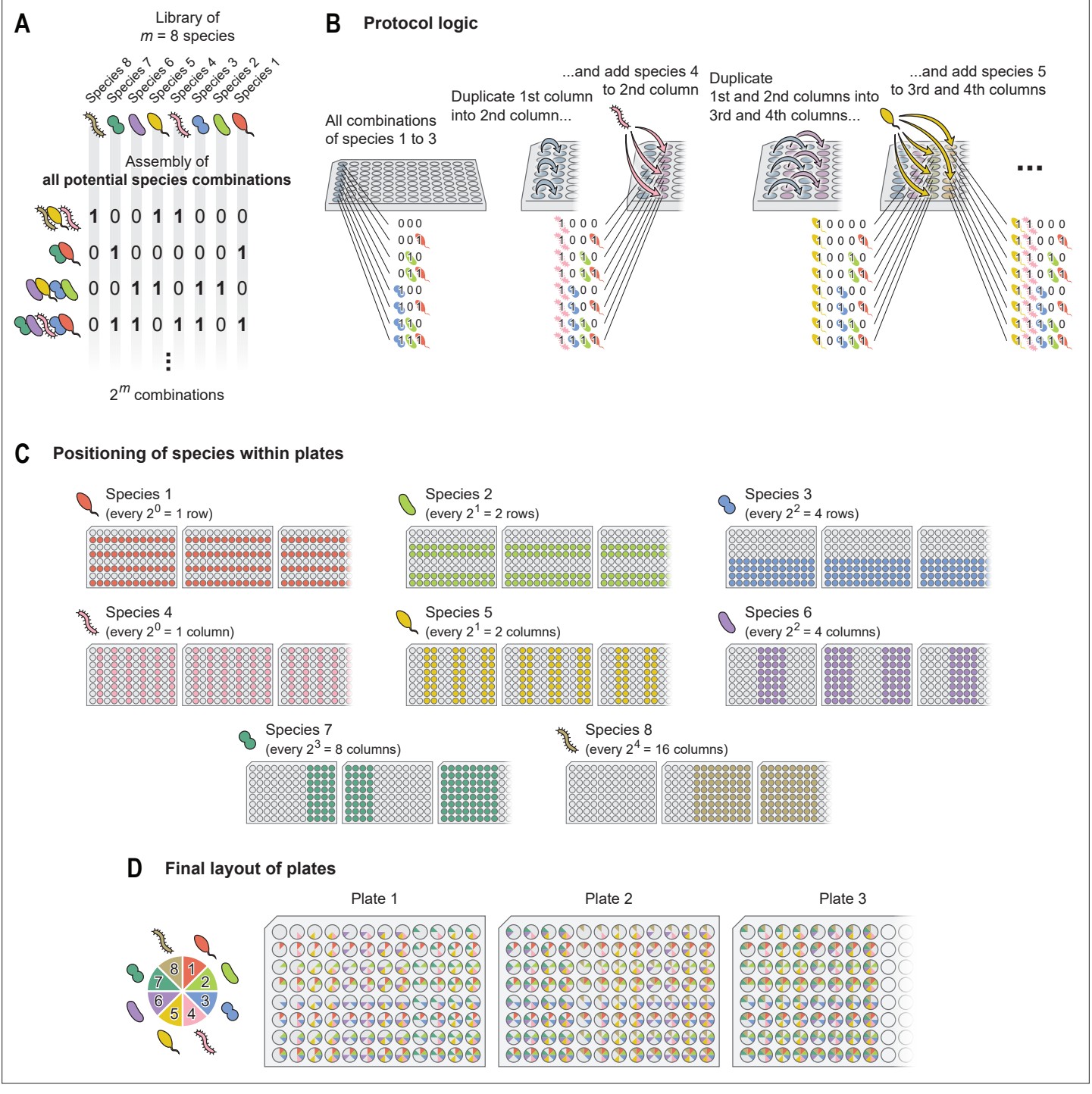

**Figure 1.** Logic of the assembly protocol. (**A**) We consider a set of $m$ species (in this illustration, $m=8$), and we identify each of the $2^m$ consortia that can be formed with them by the binary number $c=x_m x_{m-1} \cdots x_2 x_1$ (with $x_k=1,0$ representing the presence/absence of species $k$, respectively; see **Equation 1**). (**B**) Our protocol is based on the idea that one could assemble each of the $2^3=8$ combinations of species 1–3 in the first column of a 96-well plate. Iteratively duplicating these consortia and adding species 4 to $m$ to them would eventually result in all $2^m$ combinations of species, as described in the main text. (**C**) Each of the $m$ species would be present in the indicated wells (see **Equation 2**). (**D**) Positioning of the $2^m$ consortia within the 96-well plates used in the protocol.

these have 16 rows. In this case, we would use the first column to assemble all consortia made up by species 1–4, and then proceed as described above for all subsequent species.

## Plate arrangement

Following the logic we just discussed, it is straightforward to notice that species 4 will be present in all wells of columns 2, 4, 6, 8… of the 96-well plate. In turn, species 5 will be present in alternating pairs of columns: 3, 4, 7, 8, 11, 12… Species 6 will be present in alternating sets of four columns: 5, 6, 7, 8, 13, 14, 15, 16… (*Figure 1C*). Note that 96-well plates only have 12 columns: what we here call columns 13–24 would in practice be columns 1–12 of a second 96-well plate, columns 25–36 would correspond to a third plate, and so on. By a similar reasoning, species 1–3 will be present in alternating sets of 1, 2, and 4 rows (instead of columns), respectively (*Figure 1C*). In summary, we have that species $k$ will be present in:

$$\text{Species } k \text{ present in:} \begin{cases} \text{alternating sets of } 2^{k-1} \text{ rows} & \text{if } k = 1, 2, 3 \\ \text{alternating sets of } 2^{k-4} \text{ columns} & \text{if } k = 4, 5, \ldots, m \end{cases} \tag{2}$$

Conveniently, this arrangement makes it so consortia are located in order from top to bottom and from left to right within the plate(s): the first column contains the consortia corresponding to the binary representations of decimal integers 0 (00000)–7 (00111), the second column corresponds to decimal integers 8 (01000)–15 (01111), the third column to decimal integers 16 (10000)–23 (10111), and so on.

## Homogenizing species' densities across consortia

As we just saw, it would be possible to form all combinations of $m$ species simply by pipetting each one of them row-wise (for species 1–3) or column-wise (for species 4 to $m$) in their corresponding positions (*Equation 2*) with the aid of multichannel pipettes. There is, however, one important caveat: if we pipette equal volumes of each species' monocultures, we will end up with variable volumes across wells. As an example, if we pipette a volume $v_0$ of each monoculture, we will have a volume of $2v_0$ in well D01 of the 96-well plate (where consortium 011 is located), but a volume of $3v_0$ in well H01 (where consortium 111 is located). Species 1 is present in both of these example consortia at the same total population size, but at different population densities differing by a factor of 2/3.

Often, we may want the density of each species to be consistent across consortia. To achieve this, we can pipette additional liquid (sterile water, growth medium, saline buffer…) into each well in order to compensate for the differences in volume. Note that, prior to this density homogenization step, the maximum volume in a well would be $mv_0$, corresponding to the consortium with all $m$ species. This naturally also corresponds to the consortium where each of the individual species is present at the lowest population density. In every other consortium (say, consortium $c$), the volume of buffer (which we denote $v_B(c)$) that we would need to add in order to reach this minimum population density is given by

$$v_B(c) = v_0(m - H(c)) \tag{3}$$

where $H(c)$ represents the Hamming weight of the binary representation of consortium $c$. The Hamming weight of a binary number is the number of ones in its representation, for instance, $H(0000) = 0$, $H(0010) = 1$, $H(1111) = 4$, etc. Thus, $H(c)$ simply counts the number of species present in consortium $c$.

In practice, we do not need to manually pipette the buffer well-by-well. The convenient arrangement of the plate allows us to streamline this process. In what follows, we use $i$ and $j$ as the row and column indexes of the plate (from top to bottom and from left to right, respectively). For instance, well C02 (containing consortium 1010) would correspond to $i=3$ and $j=2$. As a reminder, we are considering that $j=13$ to 24 correspond to columns 1–12 of a second 96-well plate, $j=25$ to 36 correspond to a third plate, etc.

In order to streamline the pipetting of buffer, it would be convenient to find an expression for $H(c)$ in terms of the positional indexes $i$ and $j$. Now, due to the particular arrangement of consortia in the plate, we can divide the binary representation of a consortium $c$ into two. First, the right-most

three bits of $c$ are determined by the row in which it is located. For instance, all consortia in row 7 will always have a binary representation ending in 110, independently of the column in which they appear. In general, the three rightmost digits of $c$ will be equal to $B(i-1)$, the binary representation of $i$-1 with $i$ being the index of the row where consortium $c$ is located. Similarly, the leftmost $m$-3 bits of $c$ will depend only on the column, in the following way: take a column $j$ and ignore the three rightmost digits; then the binary representation of all consortia in that column will be the same, and equal to $B(j-1)$. For instance, with $m$=8, the binary representation of all consortia in column 4 is $00011x_3x_2x_1$ — the last three bits may be 0 or 1 depending on the specific well, but the first five bits are common to all wells in the column. Therefore, in order to calculate the Hamming weight of a given consortium, we just need to sum the Hamming weight of the row and the Hamming weight of the column:

$$H(c) = H(B(i-1)) + H(B(j-1)) \tag{4}$$

We can use *Equation 4* to rewrite *Equation 3* in terms of the positional indexes $i$ and $j$. After some algebra, we arrive at

$$v_B(i,j) = v_0\left[3 - H(B(i-1))\right] + v_0\left[m - 3 - H(B(j-1))\right] \tag{5}$$

While terms in this equation could be arranged in a more compact form, *Equation 5* as it is allows us to implement the buffer pipetting in a very efficient way: We can start by pipetting liquid buffer column-wise (a volume $v_0\left[m - 3 - H(B(j-1))\right]$ into every well of column $j$). We can then pipette the buffer row-wise, pipetting a volume $v_0\left[3 - H(B(i-1))\right]$ into every well of row $i$. Furthermore, the row-wise buffer pipetting can be bypassed if we homogenize the densities of the starting consortia formed with species 1–3, as we explain in detail in Appendix 1.

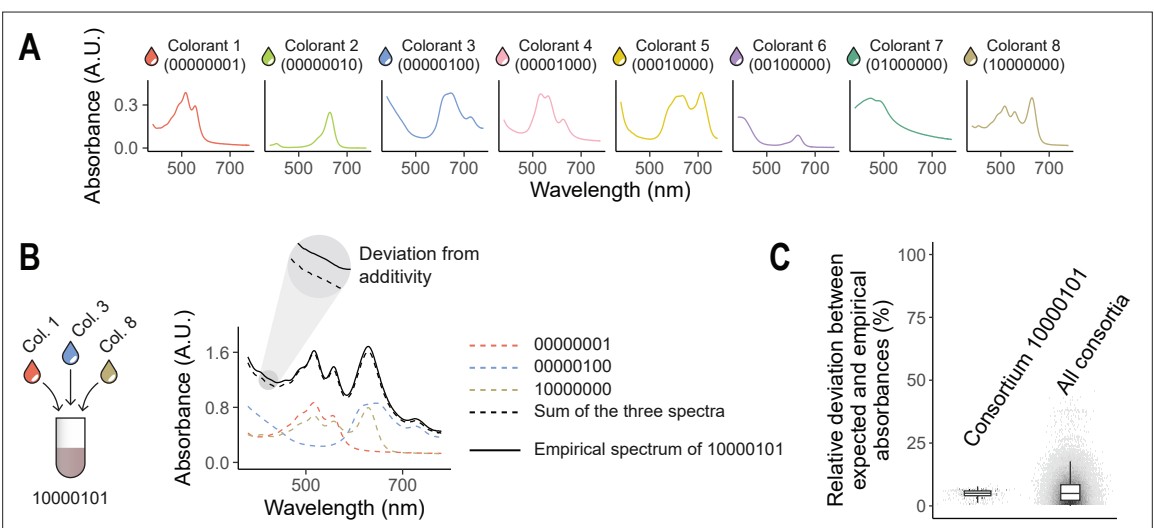

**Figure 2.** Full factorial construction of synthetic colorant combinations. (**A**) We consider a set of eight synthetic colorants, each of which exhibits a different absorbance spectrum in the range 380–780 nm. (**B**) As an example, we show the empirically measured absorbance spectrum (solid black line) for the combination of colorants 1, 3, and 8 (10000101). The dashed black line represents the additive expectation, that is the sum of the spectra of the three constituent colorants (colored dashed lines). (**C**) Relative deviation between the empirical spectrum and the additive expectation (see *Equation 6*) for the example combination 10000101 (left) and for all mixtures of two or more colorants (right). Relative errors were only computed when the true absorbance was above 0.1 A.U.

The online version of this article includes the following figure supplement(s) for figure 2:

**Figure supplement 1.** Plate layout at different stages of the assembly protocol for combinations of eight colorants.

**Figure supplement 2.** Absolute deviation, $\left|\text{Abs}(c) - Abs^{(\text{add})}(c)\right|$, between the spectra of the eight-colorant consortia and the additive expectations, for all consortia $c$ of two or more colorants and for all wavelengths $\lambda$ between 380 nm and 780 nm.

**Figure supplement 3.** Relative deviation between the empirical spectra and the respective additive expectations (*Equation 6*) versus the number of colorants in the consortium.

**Figure supplement 4.** Relative deviation between the empirical spectra and the respective additive expectations (*Equation 6*) as a function of the wavelength.

## Results

### Proof of concept: construction of colorant combinations

As a first demonstration of the method, and to establish its feasibility and accuracy, we used commercial food colorants and temperas to build all 256 combinations made up from eight colors. We diluted these colorants in water such that all of them had comparable maximum absorbances in the range from 380 nm to 780 nm. Each color exhibits a different absorbance spectrum in this range (**Figure 2A**).

Using synthetic colorants to test our protocol has various advantages. First, color differences are clearly visible to the naked eye, making it easy to keep track of the location of each colorant. Second, we do not expect these colorants to interact with one another, or to do so very weakly. This means that the absorbance spectrum of any combination of colorants may be reasonably expected to match the sum of the individual spectra of each colorant. Therefore, the magnitude of the deviations between an empirical spectrum and this additive expectation (**Figure 2B**) can be used to estimate an upper bound for the pipetting error of the protocol. We can obtain an estimation of the accuracy of our protocol by examining these deviations.

We followed the protocol described above, setting $m=8$ and $v_0=25$ µL (a detailed step-by-step protocol for these parameters is given in Appendix 1), to assemble all 256 color combinations. The absorbance spectrum of each combination was quantified using a MultiSkan SkyHigh plate reader (Thermo Fisher Scientific). The implementation of the protocol required three 96-well plates, 8 Falcon tubes, an 8-channel pipette, and a 12-channel pipette. Completing the whole protocol took a single experimentalist less than an hour. **Figure 2—figure supplement 1** shows the three 96-well plates at six different steps of the protocol.

**Figure 2B** shows the absorbance spectrum for an example colorant mixture (10000101), comparing it to the additive expectation that its three constituent colorants (1, 3, and 8) do not interact. We estimated the pipetting error of the protocol as the relative deviation (denoted $\delta$) between the empirical spectra and the additive expectations at all wavelengths (**Figure 2B**), that is:

$$\delta\left(c, \lambda\right) = \frac{\left| Abs_\lambda\left(c\right) - Abs_\lambda^{(\mathrm{add})}\left(c\right) \right|}{Abs_\lambda\left(c\right)} \tag{6}$$

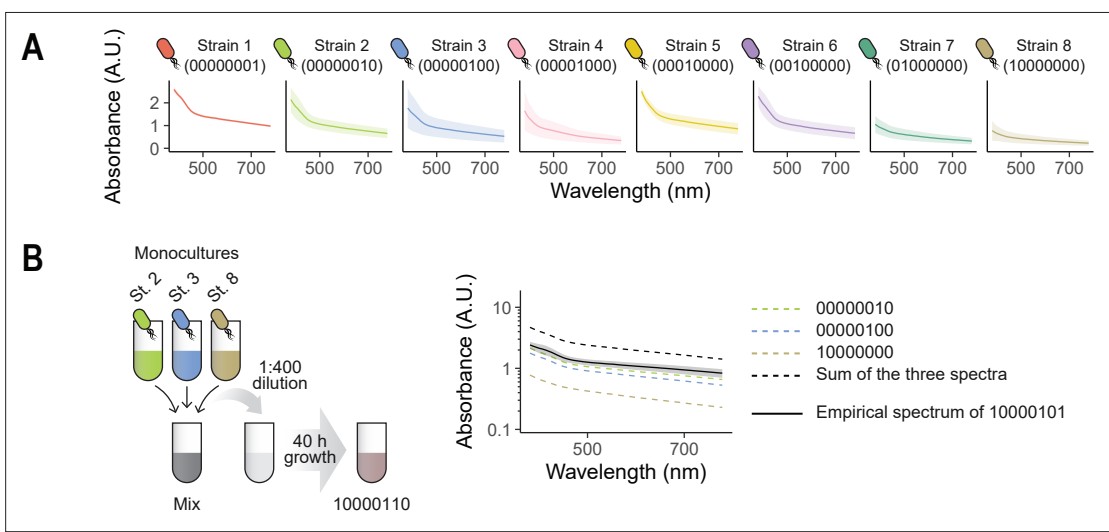

**Figure 3.** Co-culturing combinations of bacterial isolates. (**A**) We consider eight *P. aeruginosa* strains obtained from a previous experiment (**Hernando-Amado et al., 2020**; **Hernando-Amado et al., 2022a**; **Hernando-Amado et al., 2022b**). Here, we represent the absorbance spectra of the eight monocultures of these strains. Solid lines and shaded areas represent means and standard deviations across three independent biological replicates. (**B**) We followed our protocol to assemble different combinations of strains (in this example, the consortium formed by strains 2, 3, and 8, represented as 10000110). Each consortium was diluted into fresh medium by a factor of 1:400 and allowed to re-grow for 40 hr, after which the absorbance spectrum of the culture was measured. Solid black line: empirical spectrum of consortium 10000110 (mean of three independent biological replicates, shaded region represents the standard deviation across them). Dashed black line: additive expectation if the three strains did not interact. Dashed colored lines: monoculture spectra of the three constituent strains.

where we have denoted $Abs_\lambda (c)$ the absorbance at a wavelength $\lambda$ (between 380 and 780 nm) of the colorant mixture $c$, and $Abs_\lambda^{(add)} (c)$ the additive expectation (i.e. the sum of the absorbances of each individual constituent colorant).

In *Figure 2C*, we show the distribution of relative deviations for the example mixture 10000101 at all wavelengths (left), as well as the distribution corresponding to all mixtures at all wavelengths (right). *Figure 2—figure supplement 2* shows the absolute deviations for all mixtures at all wavelengths (median absolute deviation ~0.025 A.U.). We found that relative deviations were generally small (*Figure 2C*: mean ~5.8%, median ~4.9%). As a reference, we note that typical lab micropipettes have an error of 0.5–1%, and so these numbers are compatible with cumulative pipetting variation. We also found that the magnitude of these deviations did not increase with the number of colorants in a mixture (*Figure 2—figure supplement 3*), suggesting that the precision of our protocol is not compromised by the size of the set from which mixtures are assembled. Deviations were also roughly similar in magnitude across all wavelengths (*Figure 2—figure supplement 4*).

## Full factorial design of synthetic bacterial consortia

To illustrate the potential applications of our protocol for finding optimal microbial consortia and quantitatively studying the complexity of microbial interactions, we adopted a collection of eight *Pseudomonas aeruginosa* strains obtained from a previous experiment (*Hernando-Amado et al., 2020*; *Hernando-Amado et al., 2022a*; *Hernando-Amado et al., 2022b*; see Appendix for details). Monocultures of each of these strains grown in liquid LB medium exhibit differences in their absorbance spectra (*Figure 3A*). Unlike the synthetic colorants, bacterial strains can be expected to interact with one another when co-cultured through a variety of potential mechanisms, including competition for limiting resources or facilitation, which may lead to changes in the population sizes of different species (*Sanchez et al., 2023*; *Sanchez-Gorostiaga et al., 2019*).

We first cultured each strain separately by resuspending a single colony into 15 mL of LB medium in 50 mL conical Falcon tubes. Strains were allowed to grow to carrying capacity for 24 hr at 37 °C with no agitation. Then, monocultures were fully homogenized and used to assemble all 256 strain combinations following the same protocol described before (detailed steps can be found in Appendix 1). Phosphate buffer saline (1×PBS) was used to adjust volumes/densities as indicated in the protocol. After completing our protocol (which took a single experimentalist less than one hour), each consortium was inoculated into fresh LB medium (0.5 μL of inoculum into 200 μL of medium for a dilution factor of 1:400; note that our protocol design makes it so this is equivalent to inoculating 0.5 μL/8=0.0625 μL of each strain's monoculture into the fresh medium) in 96-well plates. Samples were incubated still at 37 °C for 40 hr, after which the absorbance spectrum of each sample was quantified in the same plate reader we used before.

*Figure 3B* shows the absorbance spectrum for one example consortium after the 40 hr incubation period. Consistent with our reasoning, this spectrum deviates strongly from the additive expectation (sum of each monoculture spectrum), indicating a prominent effect of interactions between strains. *Figure 4* shows the spectra for all consortia with two or more members, where deviations from the additive expectation are similarly large.

Because we have constructed all possible assemblages, we can systematically investigate the interactions between species and how they may result in different community-level properties and functions. As an illustration, we focus on the absorbance of the liquid cultures at 600 nm ($Abs_{600}$). While this is a commonly used proxy for biomass in microbial cultures, here we use it simply as an example of a quantitative community-level function in itself. Note that the relationship between absorbance and biomass is only linear at low values of $Abs_{600}$; readers interested in reliably quantifying biomass should keep this limitation in mind when designing experiments. We found a non-monotonic relationship between $Abs_{600}$ and the number of inoculated strains, peaking at three strains (*Figure 5A*). Our protocol allows us not only to characterize this type of broad-scale relationships between diversity and function, but also to identify specific high-performing consortia (*Figure 5B*). In our experiment, the consortium with the highest function contains only three strains (consortium 00001101, containing strains 1, 3, and 4). Note that the ranking depicted in *Figure 5B* serves mere illustrative purposes, as in our data many consortia exhibited similarly high biomass values. As a reference, consortium 00001101 had $Abs_{600}$=1.294 $\pm$ 0.048 A.U. (mean and standard deviation across $n$=3 biological replicates; see

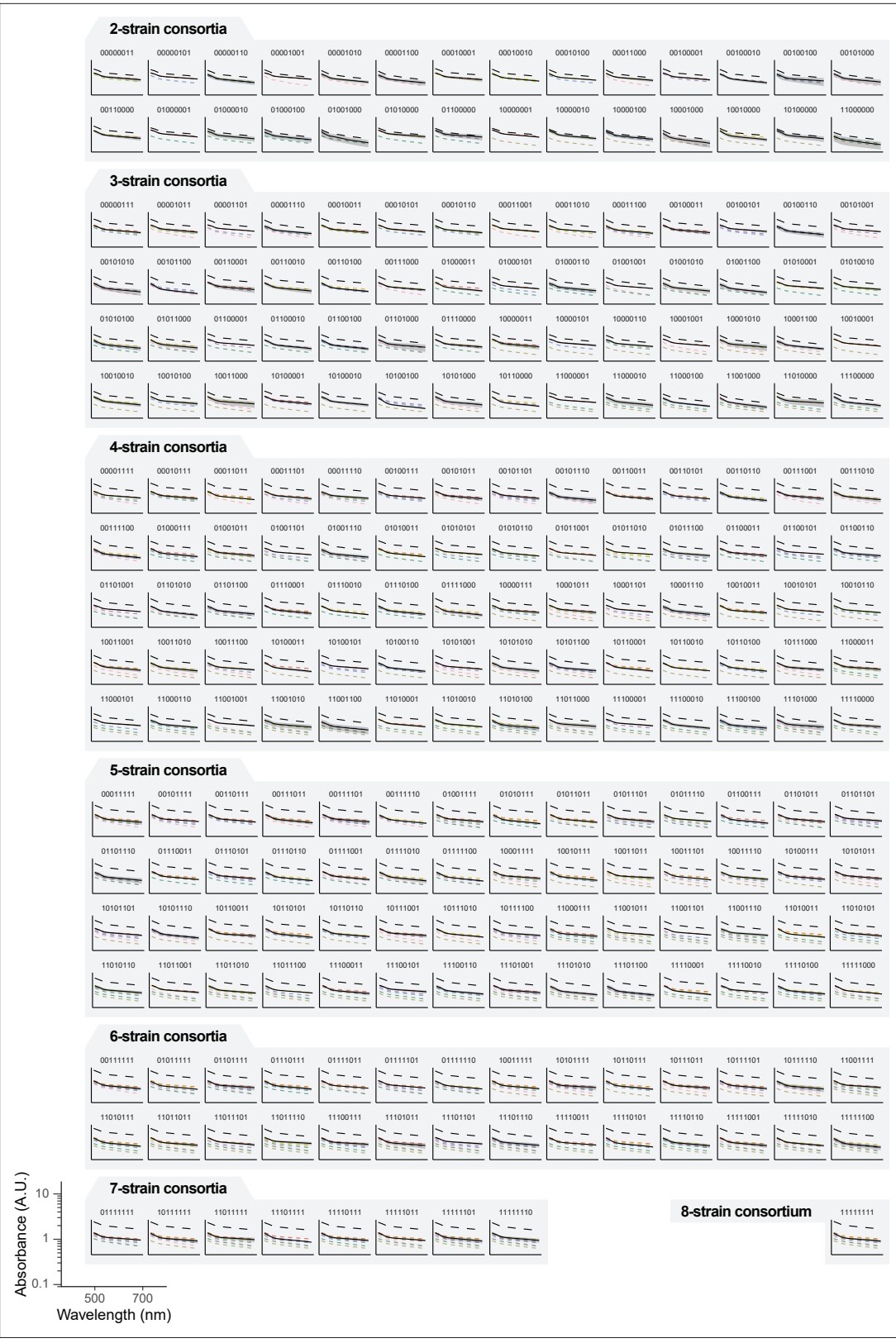

**Figure 4.** Full factorial design of synthetic microbial consortia. We represent the absorbance spectra of all consortia (of 2 or more strains) that can be formed with our library of eight *P. aeruginosa* isolates. As in *Figure 3B*, solid black lines and shaded areas represent the mean and standard deviation of the co-culture spectrum across three biological replicates; the dashed colored lines represent the individual monoculture spectra; and dashed black lines represent the sum of these individual spectra.

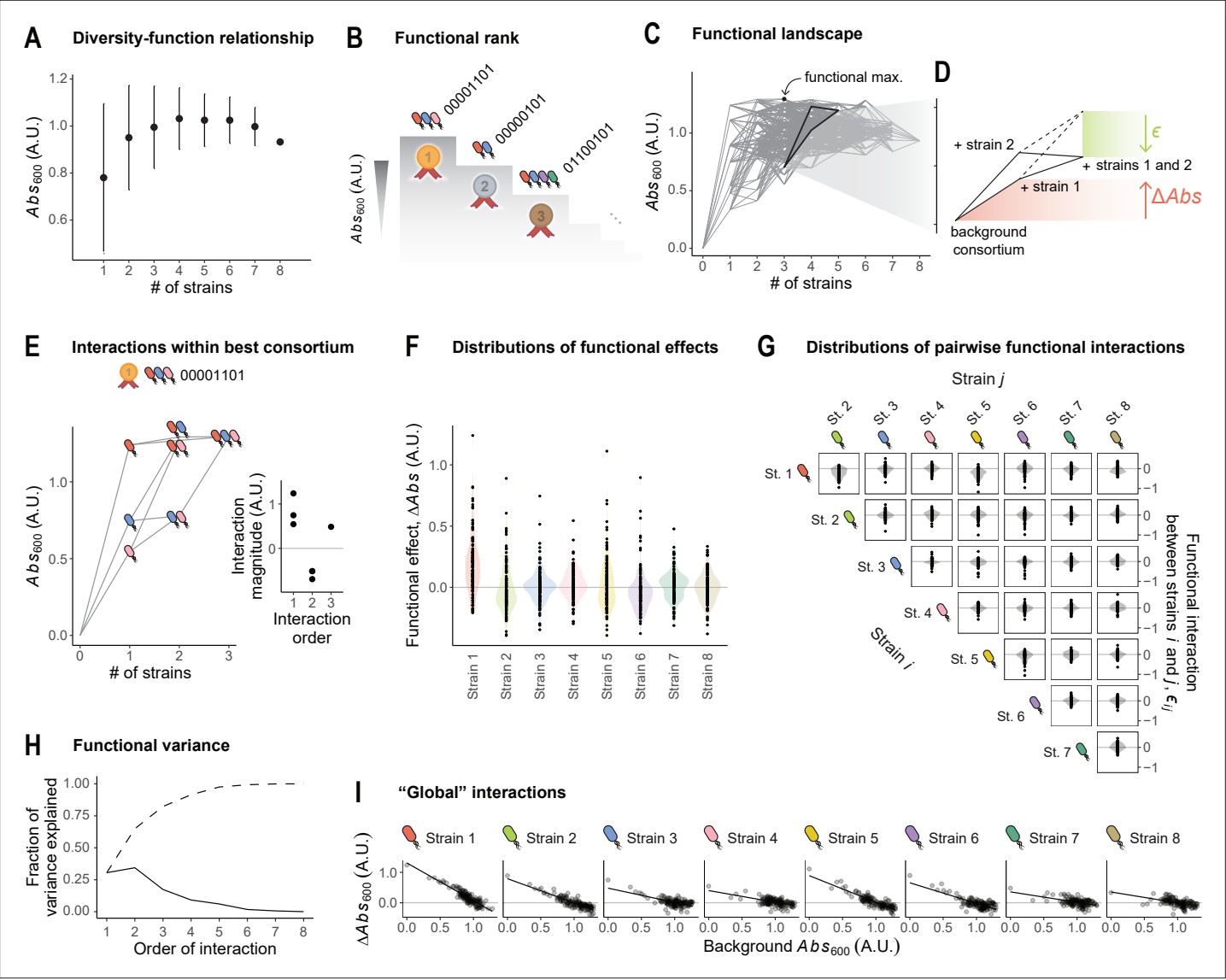

**Figure 5.** Full factorial design of microbial consortia enables the dissection of microbial interactions. We focus on the absorbance at 600 nm as a proxy for the biomass of a consortium. (**A**) Consortium function (biomass quantified as $Abs_{600}$) against the number of strains inoculated into the consortium. Dots and error bars represent the average and standard deviation across all consortia with the same number of strains. (**B**) Top three consortia exhibiting the highest biomass across all $2^8$ combinations. (**C**) We represent the function ($Abs_{600}$; values shown are averages across three biological replicates) for the consortia assembled using our protocol. Nodes correspond to the different consortia; edges connect consortia that differ in the presence of a single particular strain. Bold lines correspond to the detail highlighted in panel D. (**D**) Detail showing the effect of including strain 1, strain 2, and both, in a background consortium (in this example, the background corresponds to consortium 11001000). Red: the *functional effect* of strain 1 is quantified as the difference in function between the background consortium and the consortium resulting from the inclusion of that strain. Green: the *functional interaction* between strains 1 and 2 in this particular background is quantified as the difference between the additive expectation (dashed line) and the empirical value for the function of the consortium containing both strains, as explained in the main text. (**E**) Detail of the functional landscape corresponding to the three strains that are part of the overall highest-biomass consortium (00001101). Inset: functional interactions within this three-strain consortium. Order 1 corresponds to the monoculture functions. Order 2 corresponds to pairwise species-by-species interactions as defined in panel D of this figure. Order 3 corresponds to the third-order interaction between these strains, defined as the deviation between the empirical function of the trio with respect to the null expectation from the additive and pairwise interactions (*Sanchez-Gorostiaga et al., 2019*; *Sanchez, 2019*; *Mickalide and Kuehn, 2019*). (**F**) Distributions of functional effects of the eight strains used in our experiments across all of the 128 backgrounds where each may be included. (**G**) Distributions of functional interactions between all pairs of strains across all backgrounds where both may be included. (**H**) Fraction of functional variance due to interactions of different orders (order 1 corresponds to the additive effects of each strain). The fractions were computed as explained in Appendix 4 (*Skwara et al., 2023*). Dashed line is the cumulative sum. (**I**) We have recently shown that the functional effect of a strain is often predictable from the function of the background consortium through simple, linear regressions (*Diaz-Colunga et al., 2024*). For the eight

*Figure 5 continued*

*P. aeruginosa* strains in our experiment, linear regressions do an excellent job at linking these functional effects with the function of the background consortia.

The online version of this article includes the following figure supplement(s) for figure 5:

**Figure supplement 1.** Previous work has expressed the community-level function of a consortium *c* as $F\left(c\right)=\alpha_0+\sum_i \alpha_i\sigma_i+\sum_i \sum_j \alpha_{ij}\sigma_i\sigma_j+\sum_i \sum_j \sum_k \alpha_{ijk}\sigma_i\sigma_j\sigma_k+\cdots$ (*Skwara et al., 2023*).

**Figure supplement 2.** Correlations between the functional effect of a strain (ΔII) and the function of the background consortium where it is included, such as those we report in *Figure 5I*, can emerge when the mapping between community composition and function is entirely random (*Diaz-Colunga et al., 2024*).

Appendix 2), consortium 00000101 had $Abs_{600}=1.291 \pm 0.066$ A.U., and consortium 01100101 had $Abs_{600}=1.289 \pm 0.05$ A.U. In *Figure 5C*, we represent the full map between the composition and function of all 256 consortia in our experiment (i.e. the *community structure-function landscape Sanchez et al., 2023*; *Gould et al., 2018*; *Sanchez-Gorostiaga et al., 2019*; *Eble et al., 2023*).

From this type of data, one can readily quantify the sign and magnitude of strain interactions with respect to the community-level function of interest, in our case $Abs_{600}$. In *Figure 5D*, we show that the inclusion of strain 1 in a given consortium (which we refer to as an *ecological background*) results in an increase in function, and the same is true for the inclusion of strain 2. We call the difference in function between two consortia with and without a given focal strain the *functional effect* of that strain in that ecological background (*Diaz-Colunga et al., 2024*). If two strains did not interact, we may expect that including both of them in the same ecological background would result in a change in function equal to the sum of their two individual functional effects (see *Figure 5D*, dashed line represents this additive expectation). If the empirical function deviates from the additive expectation, the two strains engage in a *functional interaction* (denoted $\epsilon$ in *Figure 5D*) in that particular ecological background (see Appendix 3 for details; *Diaz-Colunga et al., 2024*; *Sanchez-Gorostiaga et al., 2019*). Under this definition, the term *interaction* should be interpreted in a statistical, rather than biological sense, where the null additive expectation serves as a mere minimal-assumptions baseline that is not intended to capture any particular biological mechanism (e.g. competition for nutrients).

Following this definition, one could dissect the interaction structure of all eight *P. aeruginosa* strains in our system. As an illustration, in *Figure 5E* we show the community-function landscape corresponding to all consortia that could be formed solely with strains 1, 3, and 4, that is the trio that maximizes $Abs_{600}$ (see *Figure 5B–C* indicating that the maximum $Abs_{600}$ is achieved by consortium 00001101). Although pairwise interactions are negative within this consortium, the three strains engage in a positive third-order functional interaction (see inset in *Figure 5E*) which rescues the function of the trio. This observation highlights the potential of full factorial design to interrogate and generate hypotheses with respect to the mechanisms that govern the emergence of community-level functions, particularly (but not only) in high-performing consortia.

Beyond this particular consortium, in *Figure 5F* we represent the distributions of functional effects of all eight strains in our library across all of the 128 ecological backgrounds where they may be included. We found that the functional effects of all eight *P. aeruginosa* strains exhibit large variation in sign and magnitude, indicating that the contribution of a strain to the total community function largely depends on the presence or absence of additional community members in the ecological background. This indicates a prominent role of inter-strain interactions in this system.

In *Figure 5G*, we represent the magnitude of the functional interactions between all pairs of strains *i* and *j* (denoted $\epsilon_{ij}$) across all the ecological backgrounds where the two may be included. We found that these pairwise interactions are highly variable across backgrounds. Pairwise interactions being sensitive to the presence or absence of additional community members can be interpreted as a signature of higher-order ecological interactions (HOIs; *Sanchez-Gorostiaga et al., 2019*; *Poelwijk et al., 2016*; *Letten and Stouffer, 2019*). By this definition, HOIs do exist in our particular experimental system (as evidenced by the variation in the pairwise interaction terms $\epsilon_{ij}$ shown in *Figure 5G*); however, most of the functional variance can be explained by low-order interactions (*Figure 5H*, the fraction of functional variance attributed to pairwise and higher-order interactions was computed as described elsewhere; *Skwara et al., 2023* see Appendix 4 and *Figure 5—figure supplement 1* for details). This has also been shown to be the case for other datasets of microbial community function (*Skwara et al., 2023*).

We have recently shown that the aggregate effect of pairwise and higher-order microbial interactions often results in emergent statistical patterns, which make it so the functional effect of a strain may be predictable from the function of the ecological background where that strain is included (**Diaz-Colunga et al., 2024**). The relationships between the functional effect of a strain and the function of the background community can often be captured through simple linear models, which mirror the patterns of *global epistasis* commonly reported in genetics. This phenomenon is also observed in the particular set of *P. aeruginosa* strains that we analyzed here (**Figure 5I**, $R^2$ averages 0.54), each of which exhibits a different relationship between its functional effect and the background function. Adopting the same analytical approach we used in previous work (**Diaz-Colunga et al., 2024**), we find that such strong global epistasis patterns deviate from the null expectation of a random community-function landscape (**Figure 5—figure supplement 2**).

Together, these analyses illustrate how a full factorial construction of microbial consortia may allow us both to identify optimal assemblages and to dissect how the interplay of pairwise and higher-order interactions shape community-level properties and functions. To our knowledge, the experiments we have just reported represent one of barely a handful of datasets consisting of every potential combination from a library of strains. The fact that it was done in under an hour per replicate gives a sense of the potential our design has to dramatically expand the number of empirical community-function landscapes in the literature.

## Discussion

We have introduced a rapid, simple, and inexpensive protocol for assembling all possible combinations of a given set of species using basic laboratory equipment such as a multichannel pipette and 96-well plates. Our protocol enables a single individual to construct all 256 possible consortia from a pool of eight species within an hour, and can be easily expanded to even higher-diversity libraries. Even without the aid of specialized equipment such as robotic liquid handlers, this procedure will make it possible to efficiently generate all communities with up to 10–12 species. Based on our own experience implementing the protocol, we estimate this limit based on manual labor and material constraints rather than conceptual limitations of the protocol itself. Under manual pipetting conditions, this represents a realistic upper bound that can be prepared within a single working day while minimizing handling error and variability.

For laboratories equipped with robotic liquid handlers, the protocol can be easily expanded to accommodate larger libraries, with the main limitation being the amount of plasticware that will be used. The rationale underlying our assembly protocol allows for flexible implementation depending on the specific needs of each laboratory (e.g. it is straightforward to generalize for 384-well plates). Our implementation also has the advantage of keeping pipetting error minimal and independent of consortium size, as we showed by creating all 256 color combinations from eight synthetic colorants. To facilitate automation and adaptation to the needs of other groups, we provide an accompanying R script to facilitate the generation of comprehensive step-by-step protocols for any number of species and pipetting volumes.

In our implementation, the 'density homogenization' step ensures that every species is present at the same density across all consortia that it is part of. Note, however, that this homogenization can be bypassed entirely, making it so the density of each species is variable across assemblages (specifically, inversely proportional to the number of strains in a given assemblage). Further variations in initial abundance can be explored, for instance, by treating two (or more) monocultures of the same species at different starting densities as distinct inputs of the protocol – but note that this will quickly increase the total number of assemblages that need to be tested.

We have demonstrated the utility of this protocol in synthetic microbial ecology by generating all possible communities from a collection of eight *P. aeruginosa* strains and quantifying the strength of pairwise and higher-order functional interactions between them. Our experiment illustrates how the full factorial design of microbial consortia enables the full characterization of ecological community-function landscapes, which can provide valuable insights into the emergence of community-level functions (such as, but not limited to, optical density at different wavelengths, as we studied here). The entire experiment, from spreading colonies to measuring community function, could be completed from beginning to end by a single experimentalist in less than one working week, and most of that time was taken by waiting for cell growth. In any case, this time frame will also depend on the specific

community-level function of interest, and the method used for its quantification. Here, we have focused on the absorbance of liquid cell cultures as an illustrative example of a possible readout, but the protocol is compatible with a wide range of other functional, interaction-based, or *omics*-based assays.

Although initially designed for building combinations of microbial species, our protocol can be readily applied for creating combinations of any soluble (or at least homogeneously distributed) compounds beyond microorganisms. For instance, it could be employed to generate all possible media compositions from a set of resources, all possible combinations of a set of antibiotics, toxins, or even bacteriophages, as well as mixed combinations thereof. We therefore anticipate that our methodology will find extensive applications and utility across diverse fields such as microbiology, ecology, evolution, and biotechnology.

## Acknowledgements

We thank Sara Hernando-Amado for providing the *P. aeruginosa* strains used in our experiment. A S and J D-C acknowledge support from grant PID2021-125478NAI00 funded by MCIN/AEI/10.13039/501100011033 and by "ERDF: A way of making Europe." P C acknowledges support from grants PID2022-142185NB C21 and PID2022-142185NB-C22 funded by MCIN/AEI/10.13039/501100011033 and by "ERDF: A way of making Europe." A S acknowledges funding by the European Union (ERC, ECOPROSPECTOR, 101088469). Views and opinions expressed are however those of the author(s) only and do not necessarily reflect those of the European Union or the European Research Council. Neither the European Union nor the granting authority can be held responsible for them.

## Additional information

### Funding

| Funder | Grant reference number | Author |
| --- | --- | --- |
| Spanish National Plan for Scientific and Technical Research and Innovation | PID2021-125478NAI00 | Juan Diaz-Colunga Alvaro Sanchez |
| Spanish National Plan for Scientific and Technical Research and Innovation | PID2022-142185NB-C21 | Pablo Catalan |
| Spanish National Plan for Scientific and Technical Research and Innovation | PID2022-142185NB-C22 | Pablo Catalan |
| European Research Council | 101088469 | Alvaro Sanchez |

The funders had no role in study design, data collection and interpretation, or the decision to submit the work for publication.

### Author contributions

Juan Diaz-Colunga, Conceptualization, Formal analysis, Validation, Investigation, Visualization, Methodology, Writing – original draft, Writing – review and editing; Pablo Catalan, Magdalena San Roman, Andrea Arrabal, Conceptualization, Formal analysis, Validation, Investigation, Methodology, Writing – original draft, Writing – review and editing; Alvaro Sanchez, Conceptualization, Formal analysis, Supervision, Validation, Investigation, Methodology, Writing – original draft, Project administration, Writing – review and editing

### Author ORCIDs

Juan Diaz-Colunga https://orcid.org/0000-0001-8995-4529
Pablo Catalan https://orcid.org/0000-0003-2826-4684
Magdalena San Roman https://orcid.org/0000-0002-2593-7727
Alvaro Sanchez https://orcid.org/0000-0002-2292-5608

Reviewer #1 (Public review): https://doi.org/10.7554/eLife.101906.3.sa1
Reviewer #3 (Public review): https://doi.org/10.7554/eLife.101906.3.sa2
Author response https://doi.org/10.7554/eLife.101906.3.sa3

## Additional files

### Supplementary files

MDAR checklist

Source data 1. protocol_8species_25uL.txt. An example output of the protocol generator script for eight species with a working volume of 25 µL (the default settings described throughout the manuscript).

Source data 2. colorants.txt. Original data for the experiment mixing synthetic colorants.

Source data 3. pseudo.txt. Original data for the experiment mixing *P. aeruginosa* strains.

Source code 1. protocol_generator.R. An R script to generate user-tailored protocols. Users can simply specify the desired number of species and working volume ($v_0$) at the top of the script. Running the script will then create an output .txt file with detailed step-by-step instructions.

### Data availability

Original data is provided as Supplementary files: *Source data 2 and 3*, correspond to the synthetic colorants dataset and the *P. aeruginosa* strain combinations dataset, respectively. Code for the analyses is available at https://github.com/jdiazc9/full_factorial_design (copy archived at *Díaz-Colunga, 2026*).

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

## Appendix 1

### Full protocol description for $m=8$ species

In this section, we describe our protocol for the assembly of all consortia from a $m=8$ species library in full detail. The reader will hopefully find that generalizing the protocol for larger (or smaller) library sizes is straightforward. We also provide an R script which generates complete step-by-step protocols with customizable parameters (e.g. different values of $m$).

### Initial considerations

As before, we will call $v_0$ the minimum pipetting volume throughout the protocol. It is important to keep in mind that each well of the final plates will carry a volume of $mv_0$. For $m=8$, we can choose $v_0=25$ μL, a volume which can be easily handled by standard pipettes, such that every well of the plates will carry a total volume of 200 μL by the end of the assembly protocol. In what follows, we will assume that at the start of the protocol we have enough volume of each of the $m=8$ individual species in monoculture. For some applications, it might be desirable to standardize the density of the monocultures, typically by diluting all cultures to the same target optical density. One should keep in mind that each species will be present in half of the $2^m$ consortia, that is, we will need a minimum volume of $2^{m-1}v_0$ (3.2 mL with our choices of $m$ and $v_0$) of each monoculture. It is of course recommended to have some extra volume available.

### Step 1: buffer pipetting

We start the protocol by pipetting the necessary buffer column-wise. Making this the first step of the protocol has the advantage that all liquid buffer handling can be done using the same set of pipette tips with no risk of cross-contamination. Using an eight-channel pipette, we pipette a volume of $v_0[m-3-H(B(j-1))]$ (note that this corresponds to the second term of *Equation 5*) into all wells of column $j$. We do this for every column from the first ($j=1$) to the final one ($j=2^{m-3}$, in our case $j=32$ since we have $m=8$ species). For eight species, and with $v_0=25$ μL, we have:

| Column index, $j$ | $B(j-1)$ | $H(B(j-1))$ | Buffer volume, $v_0\left[m-3-H\left(B\left(j-1\right)\right)\right]$ |
|---|---|---|---|
| 1 | 00000 | 0 | 125 μL |
| 2 | 00001 | 1 | 100 μL |
| 3 | 00010 | 1 | 100 μL |
| 4 | 00011 | 2 | 75 μL |
| 5 | 00100 | 1 | 100 μL |
| 6 | 00101 | 2 | 75 μL |
| 7 | 00110 | 2 | 75 μL |
| 8 | 00111 | 3 | 50 μL |
| 9 | 01000 | 1 | 100 μL |
| 10 | 01001 | 2 | 75 μL |
| 11 | 01010 | 2 | 75 μL |
| 12 | 01011 | 3 | 50 μL |
| 13 | 01100 | 2 | 75 μL |
| 14 | 01101 | 3 | 50 μL |
| 15 | 01110 | 3 | 50 μL |
| 16 | 01111 | 4 | 25 μL |
| 17 | 10000 | 1 | 100 μL |

*Continued on next page*

*Continued*

| Column index, $j$ | $B(j-1)$ | $H(B(j-1))$ | Buffer volume, $v_0\left[m-3-H\left(B\left(j-1\right)\right)\right]$ |
|---|---|---|---|
| 18 | 10001 | 2 | 75 µL |
| 19 | 10010 | 2 | 75 µL |
| 20 | 10011 | 3 | 50 µL |
| 21 | 10100 | 2 | 75 µL |
| 22 | 10101 | 3 | 50 µL |
| 23 | 10110 | 3 | 50 µL |
| 24 | 10111 | 4 | 25 µL |
| 25 | 11000 | 2 | 75 µL |
| 26 | 11001 | 3 | 50 µL |
| 27 | 11010 | 3 | 50 µL |
| 28 | 11011 | 4 | 25 µL |
| 29 | 11100 | 3 | 50 µL |
| 30 | 11101 | 4 | 25 µL |
| 31 | 11110 | 4 | 25 µL |
| 32 | 11111 | 5 | 0 µL |

## Step 2: species 4 to $m$

Using an eight-channel pipette, we now pipette a volume $v_0$=25 µL of the monocultures of species 4 to $m$ into all wells of their corresponding columns (see *Equation 2* and *Figure 1C*):

| Species | Volume | Columns, $j$ |
|---|---|---|
| 4 | 25 µL | 2, 4, 6, 8, 10, 12, 14, 16, 18, 20, 22, 24, 26, 28, 30, 32 (alternating columns) |
| 5 | 25 µL | 3, 4, 7, 8, 11, 12, 15, 16, 19, 20, 23, 24, 27, 28, 31, 32 (alternating sets of 2 columns) |
| 6 | 25 µL | 5, 6, 7, 8, 13, 14, 15, 16, 21, 22, 23, 24, 29, 30, 31, 32 (alternating sets of 4 columns) |
| 7 | 25 µL | 9, 10, 11, 12, 13, 14, 15, 16, 25, 26, 27, 28, 29, 30, 31, 32 (alternating sets of 8 columns) |
| 8 | 25 µL | 17, 18, 19, 20, 21, 22, 23, 24, 25, 26, 27, 28, 29, 30, 31, 32 (alternating sets of 16 columns) |

We recommend using different pipette tips for each pipetting event to avoid cross-contamination — but note that e.g. all pipetting of species 4 could be performed with the same set of tips with no risk of cross-contamination. Similarly, species 5 could be first pipetted into all necessary odd columns (which contain only buffer at this stage) with a same set of tips, and then into all necessary even columns (which contain only buffer and species 4 at this stage) with another unique set of tips.

## Step 3: combinations of species 1 to 3

We next assemble all $2^3$=8 combinations of species 1, 2, and 3. We recommend first assembling these consortia in large vessels, typically 15 mL or 50 mL Falcon tubes which we label as T1 to T8. We then pipette the following volumes of species 1, 2, 3 monocultures, and buffer into each of the tubes:

| Tube | Consortium | Buffer volume | Species 3 volume | Species 2 volume | Species 1 volume |
|------|-----------|---------------|------------------|------------------|------------------|
| T1 | 000 | 3 mL | 0 | 0 | 0 |
| T2 | 001 | 2 mL | 0 | 0 | 1 mL |
| T3 | 010 | 2 mL | 0 | 1 mL | 0 |
| T4 | 011 | 1 mL | 0 | 1 mL | 1 mL |
| T5 | 100 | 2 mL | 1 mL | 0 | 0 |
| T6 | 101 | 1 mL | 1 mL | 0 | 1 mL |
| T7 | 110 | 1 mL | 1 mL | 1 mL | 0 |
| T8 | 111 | 0 | 1 mL | 1 mL | 1 mL |

Lastly, we pipette a volume of $3v_0$=75 µL per well from tubes T1 to T8 into rows $i$=1 to 8 of the plates, respectively, which can be easily done with the aid of a 12-channel pipette. Note that this not only places species 1–3 in their corresponding positions (*Figure 1C*), but also adds the necessary amount of buffer per row (see first term of *Equation 5*). Also note that, since a volume of $3v_0$ per well is added from each tube to every row of the plates, a minimum volume of $3v_0\, 2^{m-3}$=2.4 mL per tube is required — but again, some spare volume is desirable, so here we have rounded that up to 3 mL per tube.

# Appendix 2

## Reagents and strains

For our experiment with synthetic colorants, we used a mix of food colorants (Dr. Oetker) and temperas, which were diluted in water until they all showed similar maximum absorbance values in the range 380–780 nm (*Figure 2A*).

All eight *P. aeruginosa* strains were kindly donated by Sara Hernando-Amado and are described elsewhere (*Hernando-Amado et al., 2020*; *Hernando-Amado et al., 2022a*; *Hernando-Amado et al., 2022b*). Strains 1–8 (as labeled in the main text) correspond, in order, to mutants pAOV, PA14, NfxB, ParR-CAZ, mexZ, OrfN, NfxB-CAZ, and MDR6-CAZ in the original publications. For our experiment, strains were streaked on LB agar (Condalab) plates from frozen stocks, and a single colony of each strain was resuspended in 15 mL of LB medium (Condalab; contained in 50 mL conical tubes, Falcon). Monocultures were allowed to grow for 24 hr at 37 °C, after which we used them to assemble all $2^8$ strain combinations as described in the main text. After assembling every combination, samples were diluted into fresh LB medium by a factor of 1:400 (0.5 µL of inoculum into 200 µL of fresh medium) in 96-well plates (Thermo Fisher Scientific). Plates were covered with transpirable seals (Excel Scientific), and cultures were incubated for 40 hr at 37 °C with no agitation. After incubation, plates were transferred to a MultiSkan SkyHigh plate reader (Thermo Fisher Scientific; software version: SkanIt 7.0) to obtain the absorbance spectra. Three independent biological replicates were assembled on different days by the same experimentalist. Each replicate was started from an independent clonal isolate of each strain, separately resuspended and allowed to grow in LB medium as indicated above. These independent cultures were then used as inputs for three independent executions of the protocol. Data shown in *Figures 3–5* correspond to the average across replicates.

# Appendix 3

## Quantification of pairwise functional interactions between species

Following recent work in microbial ecology (*Diaz-Colunga et al., 2024*; *Sanchez-Gorostiaga et al., 2019*; *Sanchez, 2019*; *Mickalide and Kuehn, 2019*), we define the pairwise functional interaction between two species $i$ and $j$ in a background consortium $B$ as

$$\epsilon_{ij}\left(B\right) = F\left(B+i+j\right) - \left[F\left(B\right) + \Delta F_i\left(B\right) + \Delta F_j\left(B\right)\right] \tag{A3.1}$$

where we have called $B+i$, $B+j$, and $B+i+j$ the consortia resulting from including species $i$, species $j$, or both, in the ecological background $B$. Here, $F$ generically denotes the function of a consortium, which in our case is the absorbance at 600 nm ($Abs_{600}$). The terms in brackets represent the additive expectation for the function of consortium $B+i+j$, that is, the background function $F\left(B\right)$ plus the two individual functional effects of species $i$ and $j$ in that background (denoted $\Delta F_i\left(B\right)$ and $\Delta F_j\left(B\right)$, respectively). The functional effect of species $i$ in background $B$, as explained in the main text, is simply quantified as the difference in function between consortium $B+i$ and $B$:

$$\Delta F_i\left(B\right) = F\left(B+i\right) - F\left(B\right) \tag{A3.2}$$

Thus, equation *Equation A3.1* can be written as

$$\epsilon_{ij}\left(B\right) = F\left(B+i+j\right) - F\left(B+i\right) - F\left(B+j\right) + F\left(B\right) \tag{A3.3}$$

*Figure 5F* in the main text shows the values of the interaction coefficients $\epsilon_{ij}$ for all pairs of strains $i$ and $j$ across all potential backgrounds $B$, calculated using *Equation A3.3*.

## Appendix 4

### Analysis of functional variance by interaction order

Recent analyses of already published microbial consortia *Skwara et al., 2023* used the following approach to quantifying interactions: instead of representing the presence/absence of a species $i$ with a variable $x_i = 1, 0$, one can represent it using a variable $\sigma_i = +1, -1$. The function (in our case, biomass quantified as $Abs_{600}$) of a consortium $c$ can thus be expressed as

$$Abs_{600}(c) = \alpha_0 + \sum_i \alpha_i \sigma_i + \sum_i \sum_{j>i} \alpha_{ij} \sigma_i \sigma_j + \sum_i \sum_{j>i} \sum_{k>j} \alpha_{ijk} \sigma_i \sigma_j \sigma_k + \cdots \tag{A4.1}$$

In this expression, the coefficients $\alpha_i$ represent the average difference in function between two consortia differing solely on the presence or absence of species $i$, that is, the average functional effect (as defined in the main text and in Appendix 3) of species $i$ across all potential ecological backgrounds where this species may be included (*Skwara et al., 2023*). Therefore, the coefficients $\alpha_i$ are the averages of the distributions shown in *Figure 5F*. In turn, the coefficients $\alpha_{ij}$ are proportional to the average pairwise functional interaction between species $i$ and $j$ (in the sense that we defined them in the main text, see *Figure 5D* and Appendix 3), that is the averages of the distributions in *Figure 5F*.

Using this representation has the advantage that the total functional variance (in our case, the total variance in $Abs_{600}$ across all $2^8$ consortia) can be expressed as

$$\mathrm{var}\left(Abs_{600}\right) = \sum_i \alpha_i^2 + \sum_i \sum_{j>i} \alpha_{ij}^2 + \sum_i \sum_{j>i} \sum_{k>j} \alpha_{ijk}^2 + \cdots \tag{A4.2}$$

The variance explained by the coefficients of order $k$ can be computed as $\sum \alpha_{(k)}^2$, e.g., the functional variance explained by the first-order coefficients is $\sum_i \alpha_i^2$, the variance explained by the second-order terms is $\sum_i \sum_j \alpha_{ij}^2$, and so on. This allows us to calculate the fraction of variance which can be attributed to each order, represented in *Figure 5H* in the main text.

