## [Editor Report · eLife Assessment]

This manuscript introduces a new low-cost and accessible method for assembling combinatorially complete microbial consortia using basic laboratory equipment, which is a **valuable** contribution to the field of microbial ecology and biotechnology. The evidence presented is **compelling**, demonstrating the method's effectiveness through empirical testing on both synthetic colorants and *Pseudomonas aeruginosa* strains.

---

## [Referee Report · Reviewer #1 (Public review)]

This work develops a simple, rapid, low-cost methodology for assembling combinatorially complete microbial consortia using basic laboratory equipment. The motivation behind this work is to make the study of microbial community interactions more accessible to laboratories that lack specialized equipment such as robotic liquid handlers or microfluidic devices. The method was tested on a library of *Pseudomonas aeruginosa* strains to demonstrate its practicality and effectiveness. It provided a means to explore the complex functional interactions within microbial communities and identify optimal consortia for specific functions, such as biomass production.

The primary strength of this manuscript lies in its accessibility and practicality. The method proposed by the authors allows any laboratory with standard equipment, such as multichannel pipettes and 96-well plates, to readily construct all possible combinations of microbial consortia from a given set of species. This greatly enhances access to full factorial designs, which were previously limited to labs with advanced technology.

Another strength of the manuscript is the measurement and analysis of the biomass of all possible combinations of 8 strains of *P. aeruginosa*. This analysis provides a concrete example of how the authors' new methodology can be used to identify the best-performing communities and map pairwise and higher-order functional interactions.

Notably, the authors do exceptionally well in providing a thorough description of the methodology, including detailed protocols and an R script for customizing the method to different experimental needs. This enhances the reproducibility and adaptability of the methodology, making it a valuable resource for researchers wishing to adopt this methodology.

Comments on revisions:

I thank the authors for their response. The revisions have addressed all of the issues raised in my original review, and I believe they have improved the clarity of the manuscript.

---

## [Referee Report · Reviewer #3 (Public review)]

The author developed a useful methodology for generating all combinations of multiple reagents using standard lab equipment. This methodology has clear uses in for studying of microbial ecology as they demonstrated. The methodology will likely be useful for other types of experiments that required exhaustive testing of all possible combinations of a given set of reagents (e.g., drug-drug antagonism and synergy).

The authors provided a useful R script that generates a detailed experimental protocol for building desired combination from any number of reagents. The produced document is useful and has clear instructions. The output of the computer script will be strengthened if graphical output is also provided (similar to the one provided in Figure 1C).

The authors show that the error rate of the method doesn't go up with the number of combinations using dyes (Figure 2).

The authors demonstrate the value of their methodology for studying interactions within microbial consortia by assembling all possible combinations of eight strains of *Pseudomonas aeruginosa*. The value of their methodology for this application is well founded. However, it is also unclear why specific experimental choices were made for this application. It is unclear why authors continue to show the absorbance measurements of strain assemblies over the entire wavelength spectrum and not just for ABS 600 nm (figures 3 and 4). It is also unclear why the authors provided information on the "sum of the three spectra" as this reference line is meaningless and not a reasonable null model for estimating how well specific strain combinations will grow together.

Figure 5 illustrates the various analysis types that can be performed on the data collected from growing combinations of eight *Pseudomonas aeruginosa* strains. It is a very informative figure since it provides a "roadmap" on the various ways in which the dataset produced can be explored. The information in Figure 5 and S6 will likely be very useful for a wide audience.

Comments on revisions:

We thank the author for considering the review and providing additional clarifications. The authors disagree with some of the points we raised and decided to reject some of our recommendations. All the points of disagreement are minor and clearly subjective (e.g., stylistic). Congratulations again for this elegant manuscript.

---

## [Author Response]

The following is the authors’ response to the original reviews.

**Reviewer #1 (Public review):**
This work develops a simple, rapid, low-cost methodology for assembling combinatorially complete microbial consortia using basic laboratory equipment. The motivation behind this work is to make the study of microbial community interactions more accessible to laboratories that lack specialized equipment such as robotic liquid handlers or microfluidic devices. The method was tested on a library of *Pseudomonas aeruginosa* strains to demonstrate its practicality and effectiveness. It provided a means to explore the complex functional interactions within microbial communities and identify optimal consortia for specific functions, such as biomass production.The primary strength of this manuscript lies in its accessibility and practicality. The method proposed by the authors allows any laboratory with standard equipment, such as multichannel pipettes and 96-well plates, to readily construct all possible combinations of microbial consortia from a given set of species. This greatly enhances access to full factorial designs, which were previously limited to labs with advanced technology.Another strength of the manuscript is the measurement and analysis of the biomass of all possible combinations of 8 strains of *P. aeruginosa*. This analysis provides a concrete example of how the authors' new methodology can be used to identify the best-performing communities and map pairwise and higher-order functional interactions.Notably, the authors do exceptionally well in providing a thorough description of the methodology, including detailed protocols and an R script for customizing the method to different experimental needs. This enhances the reproducibility and adaptability of the methodology, making it a valuable resource for researchers wishing to adopt this methodology.

We thank the reviewer for their thoughtful comments and positive assessment of our work. Below we detail the changes we have introduced in the manuscript to clarify issues raised by the reviewer.

While the methodology is robust and well-presented, there are some limitations that should be acknowledged more thoroughly. First, the method's scalability is an important factor. The authors indicate that it should be effective for up to 10-12 species, but there is no discussion of what sets this scale: time, amount of labor, consumables, the likelihood of error, sample volume, etc.

The 10-12 species estimation is based on our own experience implementing the protocol, and set primarily by time, labor, and consumables (as rightly pointed out by the reviewer) rather than conceptual limitations of the approach. We have added clarifications in the Discussion (lines 401-405) regarding these scalability-limiting factors.

Second, this methodology is tailored to construct communities where the abundance of each strain is identical in each combination. Therefore, combinations with a different number of strains also differ in the total initial amount of microbial cells. Second, variations in the initial proportions of the same set of strains cannot be readily explored.

Note that the “density homogenization” step is optional and it could be skipped entirely, which would result in a same species being present at variable densities across consortia: specifically, skipping this step would make the density of a species in a consortium inversely proportional to the number of species in that consortium. Further variations in initial abundance could be explored by treating a same strain at two (or more) starting abundances as distinct inputs of the protocol – though this would naturally increase the number of combinations to test.

We have included a paragraph in the Discussion (lines 416-423) describing how we can, in principle, extend our protocol to explore abundance effects.

Third, the manuscript only discusses how to construct the combinations, and not how to assay them afterward (e.g. for community function, interspecific interactions, etc.). While details on how to achieve these goals are clearly outside the scope of this work, the use of biomass as an example function may obfuscate this caveat, which should be stated more explicitly.

We agree that the manuscript focuses exclusively on the construction of microbial communities and does not address how these communities should be assayed afterward. This is an intentional scope decision. The proposed protocol is fully compatible with a wide range of functional, interaction-based, or omics-based assays. Absorbance is mentioned as an illustrative example of a possible readout, rather than as a recommended or exclusive parameter. We have revised the text to explicitly state that the assessment of community function or interspecific interactions lies outside the scope of this work and must be tailored to the specific biological question being addressed.

**Reviewer #1 (Recommendations for the authors):**
A few specific technical notes and notes about clarity:(1) It may be worth being more explicit about how to produce replicates. For example, producing technical replicates by inoculating multiple times from the same set of combinations, while biological replicates require making the combinations multiple times.

We have updated the main text to clarify this point (line 780-781).

(2) Figure 2C: May be worth adding some context to these performance numbers. What are typical accuracies? What would they be in a liquid handler?

Assessing typical accuracies is nuanced since the error depends not only on the assembly steps, but also on potential intrinsic variation of the specific community function being tested and the method used to quantify it. One of the main reasons for including the experiment using colorant combinations was precisely to minimize these other sources of variation. In this experiment, we find that the error we quantify is consistent with cumulative pipetting variation (as a reference, a typical lab micropipette has an error of 0.5-1%). This is now explicitly mentioned in the manuscript.

(3) Figure 5A: I realize it is unlikely that strains go extinct in these experiments. But it is still worth clarifying that the number of strains is the number inoculated, rather than the one present at the time of measurement.

We updated the caption of Figure 5A as recommended by the reviewer.

(4) Figure 5B: I realize this is just for illustration purposes, but you should provide more information about the magnitude of the difference in performance of these combinations and the confidence in their ranking (or variability in performance across replicates).

Following this suggestion, we have added a paragraph where we report the variation across replicates for the highest-performing consortia (lines 318-323). Indeed, while variation across replicates is small, it is enough to produce an overlap between the confidence intervals of the function of some of the highest-performing consortia. This is now explicitly acknowledged in the manuscript.

(5) Figure 5C: I believe the bold black lines indicate the combinations shown in panel D, but that is not explicitly stated.

We have updated the caption of Figure 5C.

**Reviewer #2 (Public review):**
A simple and effective method for combinatorial assembly of microbes in synthetic communities of <12 species.Overall, this manuscript is a useful contribution. The efficiency of the method and clarity of the presentation is a strength. It is well-written and easy to follow. The figures are great, the pedagogical narrative is crisp. I can imagine the method being used in lots of other contexts too.The authors could better clarify what HOIs mean. They could address challenges with assaying community function. However, neither of these “weaknesses” affects the primary goal of the paper which is methodological.

We thank the reviewer for the positive assessment. With respect to HOIs, we recognize that defining and quantifying them is a non-trivial subject within the broader field of microbial ecology (see e.g. ref. 24 within the manuscript). Since our aim with this manuscript is methodological, as the reviewer notes, here we have done our best to avoid introducing new or ambiguous definitions. For this reason, we simply adopt a definition given in previous works (including refs. 10, 19, 24, 29, 37, and 38 in the manuscript), where the context-dependence of pairwise interaction terms is taken as a signature of HOIs. With respect to the challenges in assaying community function, please see our responses below.

**Reviewer #2 (Recommendations for the authors):**
Overall, this manuscript is a useful contribution, I appreciate the authors taking the time to write it up! I have a few relatively minor comments.(1) It would be nice in the introduction to address why we might want the full factorial construction of communities in the first place. This is an especially relevant question in light of the authors' 2023 Nat E&E paper where they showed that the function of communities can often be learned even when only a fraction of all possible communities is measured. This is addressed in part in the paragraph on line 34, but I think it might be worth expanding a bit given the focus on the paper.

We sincerely appreciate the reviewer’s feedback. In fact, one of the reasons that make full factorial construction desirable is precisely to test theoretical and computational models of community function, including (but not only) the statistical models developed in our 2023 Nature E&E paper. In that work, we showed that low-order models can explain a substantial fraction of the variation in community function in previously-published datasets, but we also predict that the same models could fail under complex structures of microbial interactions (e.g., strong high-order interactions). The protocol we present here enables the empirical quantification of such interactions, making this prediction (and others) directly testable. We have included that clarification in the revised manuscript (lines 56-58).

(2) Around line 74, I think it is worth mentioning that even this elegant design will face insurmountable practical challenges (time, liquid handling operations, number of plates will explode) for full factorial design with 20, 30, 40 species or more. This is relevant for some very complex synthetic consortia that some microbiome groups are constructing (e.g. hCom2 from Huang/Fishbach groups) https://www.sciencedirect.com/science/article/pii/S0092867422009904.

We agree with the reviewer that full factorial designs become impractical for very large species pools. These limits are now more clearly mentioned in the revised manuscript. We refer the reviewer to our response to comment #1 by Reviewer 1 for further details.

(3) The binary construction is a really nice clean way to explain the protocol. Appreciate the pedagogy!

We thank the reviewer for the appreciation.

(4) In the experiment with pseudomonas strains the consortia are grown in LB. This medium will support growth to relatively high OD (>1). At these densities, the change in OD with density is almost certainly not linear with cell density, and this nonlinearity likely depends on strain identity. In this case, the assumption of additivity may not hold. As a result, some of the observed "interactions" may simply be non-linearity in the assay and not the abundance of bacteria in the communities. Of course, this does not affect the assembly protocol in any way, but it does complicate the interpretation of interactions via this assay. I think this is worth pointing out since other researchers may have to think carefully about the assay they use when constructing these synthetic consortia. I think in this methods paper it is important to emphasize this so other researchers do not mistakenly identify interactions due to issues with the assay.

We thank the reviewer for pointing out this important aspect. In our experiment, we use Abs_600_ simply as an example of a measurable community-level function. The reviewer is absolutely correct in that mapping absorbance to biomass is nuanced at large OD values, where this relationship becomes non-linear. While this is not an issue from the perspective of the protocol itself, it is indeed an important consideration for users who may want to obtain reliable quantifications of biomass. We have updated the manuscript to explicitly mention this potential issue (lines 307-313). We have also emphasized the fact that our focus on *Abs600* is strictly for illustrative purposes, and we have removed all instances where a direct mapping from *Abs600* to biomass was implied in the text.

(5) Subtle point regarding HOIs. HOI (or pairwise) statistical interactions need not quantitatively be the same as interactions in a lotka volterra sense. I realize the authors do not explicitly use the term "interaction" in an gLV model formalism but this is how the majority of readers will interpret this term. I believe it is a research question as to how pairwise gLV interactions manifest themselves in terms of functional interactions. For example, a purely pairwise LV model could easily have HOI "functional interactions" if the function is total abundance since abundances depend nonlinearly on LV interactions. I think this part of the manuscript could be confusing to readers for this reason. I think the term "functional interaction" really helps with this issue, but just asking the authors to make sure this is clear.I say this because ref: 37 is focused on HOIs in an LV sense. Here, as the authors are aware, they are computing statistical "interactions" in the sense of epistasis. Given that they are computing this epistasis averaged across all community compositions a more appropriate citation might be [https://journals.plos.org/ploscompbiol/article?id=10.1371/journal.pcbi.1004771] where the same quantity is computed in a protein context.

We thank the reviewer for pointing out this important issue. Indeed, we use the term “interaction” in a statistical sense (as the deviation of the observed community function from a null, additive expectation) rather than in a Lotka-Volterra sense. We agree that the reference suggested by the reviewer is more appropriate in this context. We have updated the reference list accordingly.

(6) Figure 5G - a little hard to see. Any way to show this data more clearly? It looks like all interactions have a mean of 0 because of the way the data are presented.

The reviewer is indeed correct in that, as defined, the interactions that we quantify are back ground dependent, and their average across backgrounds lies near zero for all species. More than an issue with the representation, we think that this is an important empirical observation: it indicates that a same species pair may interact positively or negatively depending on its ecological context. We believe that the current representation is most appropriate for making this clear, but we would be open to discussing alternatives if the reviewer had a specific suggestion in mind.

**Reviewer #3 (Public review):**
The authors developed a useful methodology for generating all combinations of multiple reagents using standard lab equipment. This methodology has clear uses for studying microbial ecology as they demonstrated. The methodology will likely be useful for other types of experiments that require exhaustive testing of all possible combinations of a given set of reagents (e.g., drug-drug antagonism and synergy).The authors provided a useful R script that generates a detailed experimental protocol for building the desired combination from any number of reagents. The produced document is useful and has clear instructions. The output of the computer script will be strengthened if graphical output is also provided (similar to the one provided in Figure 1C).The authors show that the error rate of the method doesn't go up with the number of combinations using dyes (Figure 2).The authors demonstrate the value of their methodology for studying interactions within microbial consortia by assembling all possible combinations of eight strains of *Pseudomonas aeruginosa*. The value of their methodology for this application is well-founded. However, it is also unclear why specific experimental choices were made for this application. It is unclear why authors continue to show the absorbance measurements of strain assemblies over the entire wavelength spectrum and not just for ABS 600 nm (Figures 3 and 4). It is also unclear why the authors provided information on the "sum of the three spectra" as this reference line is meaningless and not a reasonable null model for estimating how well specific strain combinations will grow together.Figure 5 illustrates the various analysis types that can be performed on the data collected from growing combinations of eight *Pseudomonas aeruginosa* strains. It is a very informative figure since it provides a "roadmap" on the various ways in which the dataset produced can be explored. The information in Figures 5 and S6 will likely be very useful for a wide audience.
**Reviewer #3 (Recommendations for the authors):**
(1) Congratulations. I think the manuscript lays out a simple and very elegant methodology that will be useful for many. While I think the method is overall well explained and rationalized, the paper can greatly benefit from further expansion of Figure 5 at the expense of Figures 3 and 4.

We thank the reviewer for their thoughtful assessment of our work. We have considered the recommendations and discuss the following points in response.

(2) Unless I am missing something, there is no reason to present data collected across the entire wavelength spectrum for microbial assemblies (Figures 3 and 4). Moreover, using the same color palette for bacterial strains (Figure 3A) and colorants (Figure 2) is highly confusing. I suggest considering using only the 600 nm wavelength for any data collected from microbial assemblies and using a very different color palette for bacteria and colorants to avoid misinterpretation of the data.

We thank the reviewer for this suggestion. Our goal with Figures 3-4 was to illustrate the convenience of the protocol and the ease with which many measurements can be performed in parallel once the combinatorial assembly has been completed. While we focus on *Abs600* for all subsequent analyses, we chose to display the full spectra in Figs. 3-4 in hopes that future studies can make use of our rich dataset to interrogate questions on microbial interactions, with the option to focus on other wavelengths (which can effectively be treated as different community-level functions in their own right; for instance, we have previously used *Abs405* as a proxy for siderophore concentration). We think there is value in Figs. 3-4 in their current form to make this clear to readers.

(3) Unlike dye absorbance, bacterial carrying capacity has an upper limit, so summing individual population absorbance as a reference line seems unjustified. If the summation of absorbance is meant to provide a "null model" for expected growth, a more suitable model should be considered (e.g., max spectra or a weighted sum of the spectra from individual members).

We agree with the reviewer that our null model is not biologically constrained, and we did not intend to imply that the additive expectation was derived from biological principles. Instead, this additive expectation should be interpreted as a simple statistical baseline with minimal assumptions. The use of an additive baseline for quantifying microbial interactions has been addressed in the literature (see, e.g., references 10, 19, 24, 29, 37, and 38), and so here we chose to conform to this convention to avoid introducing new, non-standard quantifications of pairwise and higher-order interactions. We have revised the text to make this more explicit.

(4) The R script is a valuable tool. I think that a valuable improvement will be to also generate visual representations as part of the script’s output such as the colored plates in Figure 1C that are specific to the generated protocol.

We have updated the script so that it now also outputs a table specifying the location of each consortium within the plates. We chose to make this a text, rather than a graphics output, to ensure cross-device compatibility.

(5) The discussion rightly acknowledges the potential to extend the protocol to larger libraries using liquid handlers. To facilitate this implementation, it might be beneficial to modify the script output so that the ‘volume’, ‘plate’, and ‘column’ values are tab- or comma-delimited.

We thank the reviewer for the suggestion. We have modified the output so that it is now tab-delimited.

(6) Figures 3 and 4 do not provide a lot of insight. I would suggest combining them into a single figure and using only absorbance values at 600 nm. It would also be interesting to add a histogram of these absorbance values and possibly show histograms for subgroups (e.g. all assemblies with more than 3 strains vs all assemblies with 3 or fewer strains).

With respect to Figs. 3 and 4, we refer the reviewer to our response to comment #2. With respect to the histogram/subgroups plot, we understand that this would be a slightly modified version of the current Fig. 5A, where we show means and standard deviations across all subgroups of 1 to 8 species, and so we find it unclear what this figure would add.

(7) With the recommendations of removing or reworking Figures 3 and 4, and the fact that Figure 5 is data-rich (and extremely useful), it would be beneficial to split Figure 5 and include the data shown in Figure S6 in the main figure. The analysis in Figure 6S is valuable and it might be beneficial to elevate this analysis to a primary figure and provide a detailed explanation of its rationale and methods in the main text.

We appreciate this suggestion. In our view, we find that both the text and the figures benefit from a heavy focus on the assembly protocol, as this is the main contribution of this work. While we do think it is valuable to highlight the type and amount of data that can be collected with a full factorial assembly, as well as the types of analyses that can be performed with this data, we are afraid that allocating more space to these analyses may distract readers from the methodology itself. We have therefore chosen to keep the original structure for Figs. 5 and S6.